# Association between Glutathione S-Transferases Gene Variants and COVID-19 Severity in Previously Vaccinated and Unvaccinated Polish Patients with Confirmed SARS-CoV-2 Infection

**DOI:** 10.3390/ijerph20043752

**Published:** 2023-02-20

**Authors:** Katarzyna Orlewska, Justyna Klusek, Dorota Zarębska-Michaluk, Kamila Kocańda, Ruslan Oblap, Anna Cedro, Bartosz Witczak, Jolanta Klusek, Andrzej Śliwczyński, Ewa Orlewska

**Affiliations:** 1Collegium Medicum, Jan Kochanowski University, 25-317 Kielce, Poland; 2Institute of Biology, Jan Kochanowski University, 25-406 Kielce, Poland; 3Faculty of Medicine, Lazarski University, 02-662 Warsaw, Poland; 4Central Clinical Hospital of the Ministry of Interior and Administration, 02-507 Warsaw, Poland

**Keywords:** COVID-19, SARS-CoV-2, glutathione S-transferase, oxidative stress, GSTP1

## Abstract

As the outcome of COVID-19 is associated with oxidative stress, it is highly probable that polymorphisms of genes related to oxidative stress were associated with susceptibility and severity of COVID-19. The aim of the study was to assess the association of glutathione S-transferases (GSTs) gene polymorphisms with COVID-19 severity in previously vaccinated and unvaccinated Polish patients with confirmed SARS-CoV-2 infection. A total of 92 not vaccinated and 84 vaccinated patients hospitalized due to COVID-19 were included. The WHO COVID-19 Clinical Progression Scale was used to assess COVID-19 severity. GSTs genetic polymorphisms were assessed by appropriate PCR methods. Univariable and multivariable analyses were performed, including logistic regression analysis. GSTP1 Ile/Val genotype was found to be associated with a higher risk of developing a severe form of the disease in the population of vaccinated patients with COVID-19 (OR: 2.75; *p* = 0.0398). No significant association was observed for any of the assessed GST genotypes with COVID-19 disease severity in unvaccinated patients with COVID-19. In this group of patients, BMI > 25 and serum glucose level > 99 mg% statistically significantly increased the odds towards more severe COVID-19. Our results may contribute to further understanding of risk factors of severe COVID-19 and selecting patients in need of strategies focusing on oxidative stress.

## 1. Introduction

COVID-19 manifests by a wide range of symptoms, including asymptomatic infection, a mild upper respiratory tract inflammation, and acute respiratory distress resulting in death [1]. The factors predisposing for a more severe course of COVID-19 include advanced age, male sex, and comorbidities, such as cardiovascular disease, diabetes, chronic respiratory disease cancer, or obesity [1]. However, patients without any of these characteristics can also experience a severe course of the disease, which suggests that other factors may play a role as well, e.g., individual genetic profile of a patient. A study by Singh et al. emphasizes the importance of genetic differences (polymorphisms and varying expression levels) among populations across the world as one of the reasons behind the disproportions in the incidence and mortality due to COVID-19 [2]. Various genome-wide association studies (GWAS) have identified numerous loci in the human genome associated with severity of COVID-19 [3,4,5,6]. According to GWAS findings, a genetic risk locus on chromosome 3, a 3p21.31 gene cluster, is the strongest and most consistent signal [3,4]. Meta-analyses of 11 gene variants reported statistically significant association of *ACE2* rs2285666, *ACE2* rs2074192, *ACE2* rs2106809, *TNFA* rs1800629, and *AGTR1* rs5186 with COVID-19 severity [7]. The GenOMICC (Genetics of Mortality in Critical Care) study identified 16 novel independent associations, including variants within genes which are involved in leucocyte differentiation (BCL11A), interferon signaling (IL10RB and PLSCR1), and blood-type antigen secretor status (FUT2) [8]. As oxidative stress is associated with the outcome of COVID-19 [9], it is highly probable that polymorphisms of genes related to oxidative stress were associated with susceptibility and severity of COVID-19. Among genetic factors that contribute to different activity in response to clearance of oxidative stress products among individuals are the glutathione S-transferase (GST) gene polymorphisms. *GSTT1* and *GSTM1* polymorphisms typically involve gene deletion, resulting in the absence of the protein product [10]. A common functional polymorphism of *GSTP1* involves an A-G substitution in exon 5 (SNP—single nucleotide polymorphism), resulting in the conversion of isoleucine to valine at position 105 of the amino acid chain (Ile105Val), which significantly reduces the enzymatic activity of the protein [11]. Results of two ecological studies presented a negative relationship between the mortality and case–fatality of COVID-19 and the frequency of the *GSTT1* null genotype [12] and showed that countries with higher Val105 allelic frequency reported higher prevalence of COVID-19 and mortality due to COVID-19 [13]. These findings may, in part, explain some differences in COVID-19 mortality between East Asian and European populations by the *GSTT1* polymorphism and Val105 allelic frequency. Abbas et al. found that COVID-19 patients with the GSTT1 null genotype have a higher risk of mortality and lower overall survival [14]. Studies performed in a cohort of individuals of Serbian, Caucasian origin, revealed that individuals carrying the GSTP1-Val allele variant present lower odds of developing COVID-19 [15,16]; on the other hand, individuals carrying the GSTM3-CC allele variant present higher odds of developing COVID-19 [15]. Furthermore, combined GSTP1 (rs1695 and rs1138272) and GSTPM3 genotype showed cumulative risk regarding both COVID-19 prevalence and severity [15]. On the basis of the assumption that oxidative stress plays a significant role in SARS-CoV-2 infection, the authors speculated that GST gene polymorphisms might regulate individual susceptibility towards developing a variety of clinical manifestations in COVID-19. Thus, in this study, we aimed to assess the association of GST gene polymorphisms with COVID-19 severity in previously vaccinated and unvaccinated Polish patients with confirmed SARS-CoV-2 infection. The GSTP1 Ile/Val genotype was found to be associated with higher risk of developing severe disease in the vaccinated population. Confirmation of the research hypothesis and demonstration of the effect of the patient’s individual genetic profile on the severity of the disease could not only help to develop methods of prioritizing patients but also play a role in setting prophylactic measures in Caucasian populations.

According to “The Thousand Polish Genomes” project, the Polish cohort is homogenous and clustered within the European population. In a PCA of the European subpopulations, virtually all Polish samples (1065 out of 1076) were clustered with other Central European ancestries. The admixture analysis indicated that the Polish population is dominated by a single ancestral component, albeit at a lower percentage [17].

As the Polish population is genetically clustered in the European population, allele frequency for the European population has been used as a reference. Our results show that allele frequency in our population does not differ from the European population according to NCBI database. Namely, the allele frequency for studied SNP in the European population is A = 0.6739 and G = 0.3260, and in the Polish study cohort, it is A = 0.6818 and G = 0.3182 [18].

## 2. Materials and Methods

The study group comprised 176 patients with a confirmed SARS-CoV-2 infection. The following inclusion criteria were applied: providing an informed consent to participate in the study and a positive PCR test for SARS-CoV-2 RNA. The exclusion criteria included pregnancy during the COVID-19 infection and the patient’s death prior to the sample collection. The study group was recruited prospectively from the patients hospitalized at the Department of Infectious Diseases at the Regional Hospital in Kielce due to COVID-19 infection during a period of 4 months (December 2021–March 2022). All subjects signed written consent forms for genetic testing of *GST* genes. Afterwards, blood samples (test tubes with EDTA) were drawn by a qualified nurse. The prior appropriately coded samples were then kept frozen at −20 °C until the time of genetic testing. All demographic and clinical data of patients were collected from hospital records with the help of an expert clinician. We collected data on disease severity for patients hospitalized with COVID-19. The severity was classified according to the highest ordinal level that the patient experienced during hospitalization, using the World Health Organization COVID-19 Clinical Progression Scale. In this analysis, patients were categorized into the following groups: hospitalized without supplemental oxygen (mild disease); hospitalized with standard supplemental oxygen (severe disease), and hospitalized with invasive mechanical ventilation and additional organ support (very severe disease).

The study was approved by the Bioethics Committee of Jan Kochanowski University in Kielce, Poland (nr 82/2021).

### 2.1. Genotyping

Peripheral blood leukocytes were used as the material for genetic testing. The automatic nucleic acid extractor and genomic DNA whole blood kit (Magcore^®^, RBC BioScience, Forrestdale, WA, USA) were applied to extract the genomic DNA from the analyzed blood samples. The purity and concentration of the isolated DNA were evaluated spectrophotometrically at 260 nm and 280 nm (Denovix, DS-11). Analysis of the SNP (rs1695) polymorphism of the GSTP1 gene was conducted using the TaqMan qPCR method—endpoint genotyping (Assay ID C_3237198_20). The deletion of copies of genes GSTT1 (Assay ID Hs00010004_cn) and GSTM1 (Assay ID Hs02575461_cn) was analyzed using the qPCR relative quantification method with the TERT control gene. The Rotor-Gene instrument by Qiagen was used in all cases. The PCR amplification used approximately 10 ng of genomic DNA and was performed with an initial step of 95 °C for 10 min, followed by 50 cycles of 95 °C for 15 s (denaturation step) and 60 °C for 90 s (annealing and elongation step).

### 2.2. Statistical Analysis

Quantitative data were described by means, standard deviations, medians, quartiles, and minimum-to-maximum range. Categorical data were summarized by frequencies and percentages. Group comparisons were performed using the chi-square or Fisher’s exact test for categorical variables, the Mann–Whitney test for quantitative, non-normally distributed variables, or the *t*-test for quantitative normally distributed variables (normality of distribution was checked with the Shapiro–Wilk test). Univariable and multivariable analyses were performed, including logistic regression analysis. The statistical tests were two-tailed, and a *p*-value of <0.05 was considered statistically significant. All statistical analyses were performed using the R statistical package version 4.0.3. (The R Foundation for Statistical Computing, Vienna, Austria).

## 3. Results

The population for analysis included 176 patients admitted to the hospital due to COVID-19. Patients were classified into two groups on the basis of vaccination status: not vaccinated against COVID-19 (*n* = 92) and vaccinated against COVID-19 (*n* = 84, of whom 21 persons had received 3 doses of an mRNA vaccine). Baseline characteristics of both groups are summarized in Table 1. Vaccinated patients with COVID-19 tended to be older (mean age 70.4 vs. 65.8; *p* = 0.0157), and 38.6% of vaccinated individuals had serum glucose level > 99 mg% in comparison with 55.4% persons in the unvaccinated group (*p* = 0.0255). Severity of COVID-19 based on the WHO clinical progression scale was substantially lower for vaccinated than unvaccinated cases (*p* = 0.0001).

Univariable regression analysis showed that in the group of unvaccinated patients, BMI > 25 and glucose level > 99 mg/dL were independent variables that statistically significantly increased the risk of developing serious illness (Table 2). GSTM1, GSTT1, and GSTP1 gene polymorphism was not associated with severity of illness even after adjustment for age >70 years, BMI > 25, serum glucose level > 99 mg%, and comorbidities.

Table 3 describes results of the logistic regression analysis of the risk factors for moderate to severe forms of COVID-19 in vaccinated patients. In this population, carriers of GSTP1 Ile/Val genotype had 2.74 higher odds of developing a moderate to severe form of the disease than their respective comparators (*p* = 0.0398). The risk association remained statistically significant even after adjustment for age >70 years, BMI > 25, serum glucose level > 99 mg%, and comorbidities (adjusted OR = 3.086; 95% CI: 1.164-8.585; *p* = 0.0262).

In the entire cohort of 176 patients, vaccination statistically significantly decreased, while BMI > 25 and glucose level > 99 mg/dL statistically significantly increased, the risk of developing serious illness (Table 4). GSTM1, GSTT1, and GSTP1 gene polymorphisms were not associated with severity of illness. In the model that included vaccination and GSTP1, there was no statistically significant interaction among these variables (*p* = 0.28).

## 4. Discussion

Among three GST polymorphisms analysed in our study, the GSTP1 Ile/Val genotype was found to be associated with higher risk of developing a severe form of the disease in the population of vaccinated patients with COVID-19. No significant association was observed for any of the assessed GST genotypes with COVID-19 disease severity in unvaccinated patients with COVID-19. In this group of patients, BMI > 25 and serum glucose level > 99 mg% statistically significantly increased the odds towards more severe COVID-19.

The association of GSTP1 polymorphisms with COVID-19 severity seems biologically plausible since GSTP1 is the most commonly expressed GST gene polymorphism outside the liver, with main expression in the heart, lung, and brain [19]. GSTP1 constitutes more than 90% of the GST activity regarding chloro-2,4-dinitrobenzene (CDNB) in the lungs [19]. Moreover, GSTP can bind to JNK, inhibiting the kinase activity and protecting cells against H_2_O_2_-induced cell death. GSTP1 enhances S-glutathionylation reactions following oxidative and nitrosative stress both in vitro and in vivo [20]. In vivo endogenous GSTP negatively regulates excessive inflammatory response and inhibits sepsis-related organ dysfunction and even death, suggesting that GSTP may have a protective role in inflammation [21]. The GSTP1 Ile105Val variant (rs1695) is one of the most studied polymorphisms linked to chronic lung diseases, but the studies reported conflicting results with respect to the associations between GSTP polymorphism and the risk for COPD [22,23,24] and asthma [25,26,27]. The discrepancy in outcomes may be attributable to differences in ethnicity, age, urbanization, and smoking behavior, as well as to the small sample size [19].

The data on the role of GSTM1, GSTT1, and GSTP1 gene polymorphisms in SARS-CoV-2 infection are scarce and inconclusive. According to one ecological study, a significant negative correlation between the frequency of GSTT1 null genotype and three epidemiological COVID-19 parameters was reported—prevalence, mortality, and fatality—while the correlation between the frequency of the GSTM1 null genotype and the above-mentioned epidemiological variables was reported to be not significant [12]. After adjusting for possible confounders, the prevalence of COVID-19 was no longer associated with the frequency of the GSTT1 null genotype, contrary to mortality and fatality, which showed statistically significant negative associations with the GSTT1 null genotype [12]. Abbas et al. found that in the North Indian population, GSTM1/GSTT1 polymorphism was not shown to have a significant association with the severity of COVID-19, but death was significantly higher in patients with the GSTT1 null genotype, and patients having GSTM1 wild/GSTT1 null genotypes showed a poor survival rate [14]. In the Serbian population, no association between GSTM1 and GSTT1 genotype and COVID-19 development was observed [15]. Concerning GSTP1 polymorphism, ecologic study indicates that countries with higher Val105 allelic frequency of the rs1695 polymorphism showed higher prevalence and mortality of COVID-19 [13]. On the other hand, case–control study performed in a Caucasian population with Serbian origin revealed that individuals with heterozygous GSTP1 IleVal rs1695 genotype and individuals with at least one GSTP1* Val allele rs1138272 are less prone to develop COVID-19 [15]. Haplotype analysis of GSTP1 genotypes revealed that carriers of H2 haplotype (presence of GSTP1 rs1695 variant allele and GSTP1 rs1138272 referent allele) exhibited the lowest risk of proneness to COVID-19 [16]. Combined GSTP1 (rs1695 and rs1138272) and GSTM3 genotype exhibited cumulative risk with regard to both COVID-19 occurrence and severity [15]. As our study addressed the association of GST genes polymorphism and COVID-19 severity in previously vaccinated and not vaccinated patients hospitalized due to COVID-19, it is difficult to compare our findings with results reported so far. For example, in the studies that investigated the effect of GST genotypes in terms of COVID-19 severity, there was no information regarding vaccination status of patients with COVID-19 [14,15]. Moreover, in the Serbian study, there was no detailed description of the COVID-19 stages classified by the authors as mild and severe [15]. The distinction drawn in our study between previously vaccinated and not vaccinated patients with COVID-19 shines an interesting light on the role of GST gene polymorphisms in the development of COVID-19. In addition to confirming that the course of COVID-19 is statistically significantly milder in vaccinated patients than in unvaccinated patients in real-life clinical practice, we also observed that factors associated with the severity of COVID-19 differed in the study groups, and only in previously vaccinated patients was the GSTP1 Ile/Val genotype found to be associated with higher risk of developing a severe form of the disease.

Our study is limited by the relatively small sample size and lack of inclusion of other known oxidative-stress-related genetic variants. Moreover, in our study, it was not possible to detect mutations in SARS-CoV-2, so their potential impact on the severity of symptoms was not considered. We also cannot entirely rule out other confounding factors, although regression analysis has been used to minimize the impact of non-genetic factors on the results of our study.

## 5. Conclusions

Our results on the association between GSTP1 polymorphisms and severity of clinical manifestations of COVID-19 in previously vaccinated patients may contribute to further understanding of risk factors of developing a severe form of the disease and may be useful in better selection of COVID-19 patients who need specific pharmacological strategies focusing also on oxidative stress.

## Figures and Tables

**Table 1 ijerph-20-03752-t001:** Baseline characteristics of patients included in the study.

	Unvaccinated Patients *n* = 92	Vaccinated Patients *n* = 84	*p*-Value
Gender, *n* (%)			0.7093
Women	42 (45.7%)	36 (42.9%)	
Men	50 (54.3%)	48 (57.1%)	
Age (years)			0.0157
Mean (SD)	65.8 (15.5)	70.4 (14.5)	
Median (Q1, Q3)	64.5 (56.8, 78.0)	73.0 (64.8, 81.2)	
Range	25.0–94.0	26.0–92.5	
Age > 70 years, *n* (%)	35 (38.0%)	52 (61.9%)	0.0016
BMI > 25, *n* (%)	58 (63.0%)	50 (59.5%)	0.6319
Smoking, *n* (%)			0.4989
Missing data	1	4	
Never	49 (53.8%)	36 (45.0%)	
Former	37 (40.7%)	38 (47.5%)	
Ever	5 (5.5%)	6 (7.5%)	
Vitamin D3 > 30 ng/mL, *n* (%)	28 (30.4%)	28 (33.3%)	0.6801
Glucose > 99 mg%	51 (55.4%)	32 (38.6%)	0.0255
Comorbidities, *n* (%)	67 (72.8%)	68 (81.0%)	0.2027
Number of comorbidities			0.1528
Mean (SD)	1.5 (1.3)	1.8 (1.4)	
Median (Q1, Q3)	1.0 (0.0, 2.0)	2.0 (1.0, 2.2)	
Range	0.0–5.0	0.0–5.0	
GSTP1			0.3618
Wild type	45 (48.9%)	38 (45.2%)	
Ile/Val	40 (43.5%)	34 (40.5%)	
Val/Val	7 (7.6%)	12 (14.3%)	
GSTM1			0.8638
Wild type	47 (51.1%)	44 (52.4%)	
Null	45 (48.9%)	40 (47.6%)	
GSTT			0.4996
Wild type	79 (85.9%)	69 (82.1%)	
Null	13 (14.1%)	15 (17.9%)	
COVID-19 severity, *n* (%)			0.0001
Hospitalized without supplemental oxygen	24 (26.1%)	48 (57.1%)	
Hospitalized with standard supplemental oxygen	65 (70.7%)	34 (40.5%)	
Hospitalized with invasive mechanical ventilation and additional organ support	3 (3.3%)	2 (2.4%)	

No significant differences between groups were found regarding other evaluated parameters, such as sex, BMI, comorbidities, vitamin D3 level, smoking status, GSTT1, GSTM1, or GSTP1 distribution. GSTP1 genotype distribution of participants did not deviate from the Hardy–Weinberg equilibrium (*p* = 0.6423 for unvaccinated group, *p* = 0.3373 for vaccinated group).

**Table 2 ijerph-20-03752-t002:** Logistic regression analysis of the risk factors of mild to severe COVID-19 in unvaccinated individuals developing COVID-19.

	OR (95% CI)	*p*-Value
Gender		
Men (ref.)		
Women	0.63 (0.25–1.6)	0.3318
Age	1.01 (0.98–1.04)	0.5581
Age > 70 years		
No (ref.)		
Yes	1.7 (0.62–4.64)	0.3003
BMI > 25		
No (ref.)		
Yes	4.3 (1.61–11.47)	0.0036
Smoking		
Never (ref.)		
Former	0.61 (0.23–1.63)	0.3214
Ever	0.17 (0.03–1.17)	0.0713
Vitamin D3 > 30 ng/mL		
No (ref.)		
Yes	1.09 (0.39–3.01)	0.8752
Glucose > 99 mg%		
No (ref.)		
Yes	3.44 (1.29–9.18)	0.0136
Comorbidities		
No (ref.)		
Yes	1.14 (0.41–3.21)	0.7986
Number of comorbidities	1.17 (0.81–1.69)	0.4
GSTP1		
Wild type (ref.)		
Ile/Val	1.25 (0.46–3.38) 1.117 (0.369–3.403) *	0.6937 0.8435 *
Val/Val	0.48 (0.09–2.49) 0.401 (0.055–3.023) *	0.3859 0.3598 *
GSTM1		
Wild type (ref.)		
Null	0.94 (0.37–2.39) 1.069 (0.361-3.211) *	0.9014 0.9037 *
GSTT1		
Wild type (ref.)		
Null	1.21 (0.3–4.81) 0.732 (0.174–3.771) *	0.7899 0.6822 *

* adjusted for age >70 years, BMI > 25, serum glucose level >99 mg%, and comorbidities.

**Table 3 ijerph-20-03752-t003:** Logistic regression analysis of the risk factors of mild to severe COVID-19 in vaccinated individuals developing COVID-19.

	OR (95% CI)	*p*-Value
Gender		
Men (ref.)		
Women	0.5 (0.2–1.22)	0.1289
Age	1.02 (0.99–1.05)	0.3094
Age > 70 years		
No (ref.)		
Yes	0.77 (0.32–1.87)	0.5597
BMI > 25		
No (ref.)		
Yes	1.12 (0.46–2.71)	0.7975
Smoking		
Never (ref.)		
Former	1.13 (0.45–2.85)	0.7899
Ever	0.7 (0.11–4.33)	0.7012
Vitamin D3 > 30 ng/mL		
No (ref.)		
Yes	1.24 (0.5–3.1)	0.6402
Glucose > 99 mg%		
No (ref.)		
Yes	0.83 (0.34–2.04)	0.6891
Comorbidities		
No (ref.)		
Yes	0.7 (0.23–2.09)	0.522
Number of comorbidities	1.22 (0.88–1.69)	0.2236
GSTP1		
Wild type (ref.)		
Ile/Val	2.74 (1.05–7.18) 3.086 (1.164–8.685) *	0.0398 0.0262 *
Val/Val	1.55 (0.41–5.89) 1.463 (0.356–5.728) *	0.5217 0.5853 *
GSTM1		
Wild type (ref.)		
Null	0.97 (0.41–2.310) 0.909 (0.373–2.198) *	0.9497 0.8318 *
GSTT1		
Wild type (ref.)		
Null	0.27 (0.07–1.05) 0.258 (0.065–1.022) *	0.0593 0.0538 *

* adjusted for age >70 years, BMI > 25, serum glucose level >99 mg%, and comorbidities.

**Table 4 ijerph-20-03752-t004:** Logistic regression analysis of the risk factors of mild to severe COVID-19 in the entire cohort of individuals developing COVID-19 (vaccinated and unvaccinated).

	OR (95% CI)	*p*-Value
Gender		
Men (ref.)		
Women	0.62 (0.34–1.13)	0.1171
Age	1.0 (0.98–1.02)	0.6678
Age > 70 years		
No (ref.)		
Yes	0.8 (0.44–1.46)	0.4603
BMI > 25		
No (ref.)		
Yes	2.04 (1.1–3.78)	0.0246
Smoking		
Never (ref.)		
Former	0.77 (0.41–1.46)	0.4238
Ever	0.33 (0.09–1.21)	0.0942
Vitamin D3 > 30 ng/mL		
No (ref.)		
Yes	1.1 (0.58–2.11)	0.7648
Glucose > 99 mg%		
No (ref.)		
Yes	1.9 (1.03–3.52)	0.0406
Comorbidities		
No (ref.)		
Yes	0.79 (0.38–1.62)	0.5207
Number of comorbidities	1.12 (0.89–1.4)	0.3311
COVID-19 vaccination		
No (ref.)		
Yes	0.26 (0.14–0.5)	<0.0001
GSTP1		
Wild type (ref.)		
Ile/Val	1.76 (0.92–3.37)	0.0888
Val/Val	0.76 (0.28–2.06)	0.5902
GSTM1		
Wild type (ref.)		
Null	0.98 (0.54–1.79)	0.9444
GSTT1		
Wild type (ref.)		
Null	0.54 (0.24–1.22)	0.1409

## Data Availability

All data are available on reasonable request.

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
