# Peer review of "Association between Glutathione S-Transferases Gene Variants and COVID-19 Severity in Previously Vaccinated and Unvaccinated Polish Patients with Confirmed SARS-CoV-2 Infection"

_ijerph, 2023, doi:10.3390/ijerph20043752_

Round 1
Reviewer 1 Report
The author conducted an interesting study on discovering the association between genetic polymorphisms in glutathione S-transferases genes and COVID severity by vaccine status. To improve the study and manuscript, following comments and concerns should be addressed.
1) I would suggest revising the title to make it more clear that you are going to look at the genetic polymorphisms rather than other forms of polymorphisms. In other words, this study is basically a gene/region wide genetic association study.
2) As this study was conducted in Poland, can you please provide the genetic ancestry estimate or PCA rather than self-identified race/ethnicity? I understand that it may be difficult to obtain accurate genetic ancestry based on limited gene region, but it will be helpful to show this.
3) Can you please provide the allele frequency by different race/ethnicity of the SNPs included in these region using the public database?
4) The authors conducted stratified analyses in vaccinated vs. un-vaccinated. Analyses based on the whole cohort should be conducted too, with vaccination status included as a covariate.
5) In addition, the interaction term between vaccination status and Gene should be included to discover the effect modification on severity.
Reviewer 2 Report
This manuscript is well designed and written. It provides a new knowledge about GST polymorphism after Covid vaccination which makes it interesting. They found that GST polymorphism Ile/Val was associated with severity of disease in vaccinated people with COVID19. The topic is original and relevant in the field. Previous studies are not providing sufficient information as this one. The conclusions are consistent with the evidence. Also, the references are appropriate.
Specific improvements which should be considered by the authors regarding the methodology: if they used positive and negative control like other respiratory viral infection not covid, that would be great.
Tables are fine but if they provided some figure that would be more impressive.
Reviewer 3 Report
The manuscript entitled “Association between glutathione S-transferases genes polymorphisms and COVID‐19 severity in previously vaccinated and unvaccinated Polish patients with confirmed SARS-CoV-2 infection” by Katarzyna Orlewska et al is aimed at studying the correlation between GST gene polymorphism and COVID-19 severity. The authors describe a correlation between GSTP1 variant and COVID-19 in vaccinated patients, which makes it tempting to speculate for the existence of unbalanced immune response in these patients.
Altogether the data are well presented, the results are interesting and discussed properly. I would recommend this manuscript for publication with few minor points to consider:
1. Line 4: the term “by the” is repeated
2. Line 72: please change Cov2 to CoV2
3. I would consider adding the main finding at the end of the introduction
Good luck
Round 2
Reviewer 1 Report
The concerns have been addressed and thank you for your revision.